# Enhancing Stability of Cu/ZnO Catalysts in the CO_2_ Hydrogenation to Methanol by the Addition of MoO_3_ and ReO_3_ Promoters

**DOI:** 10.3390/nano15221730

**Published:** 2025-11-17

**Authors:** Jose Soriano Rodríguez, José Manuel López Nieto, Enrique Rodriguez-Castellón, Antonia Infantes, Daviel Gómez, Patricia Concepción

**Affiliations:** 1Instituto de Tecnología Química, Universitat Politècnica de València-Consejo Superior de Investigaciones Científicas (UPV-CSIC), Avenida de los Naranjos s/n, 46022 Valencia, Spain; josorod2@itq.upv.es (J.S.R.); jmlopez@itq.upv.es (J.M.L.N.); dgomaco@itq.upv.es (D.G.); 2Departamento de Química Inorgánica, Facultad de Ciencias, Cristalografía y Mineralogía, Instituto Interuniversitario de Investigación en Biorrefinerías I3B, Universidad de Málaga, Campus de Teatinos, 29071 Málaga, Spain; castellon@uma.es (E.R.-C.); ainfantes@uma.es (A.I.)

**Keywords:** methanol, copper, promoter, molybdenum, rhenium, CO_2_

## Abstract

The catalytic hydrogenation of CO_2_ to methanol represents a promising route for carbon recycling and hydrogen storage. However, the stability of current catalysts remains one of the main technological challenges. In this work, we investigate the promotional effect of MoO_3_ and ReO_3_ on Cu/ZnO-based catalysts with metal loadings ranging from 0.06 to 3.5 wt%. Spectroscopic (XPS and in situ Raman) and kinetic studies reveal that the incorporation of ultra-low promoter amounts (0.06 wt%) enhances methanol productivity, whereas higher concentrations lead to partial blocking of the active copper sites. Rhenium promotes the stabilization of Cu^+^ species, while molybdenum establishes strong Cu-Mo interactions that modify the reducibility and surface composition of the catalyst. Remarkably, long-term stability tests (80 h, 240 °C, 20 bar and CO_2_/H_2_ = 3) demonstrate that Mo-promoted catalysts exhibit superior durability, reducing the deactivation constants by up to 82% compared to the un-promoted Cu/ZnO sample. This enhanced stability is attributed to the higher Cu-MoO_3_ interaction, enhanced Cu dispersion and high water affinity of Mo species, which trap water as Mo-OH bonds, preventing copper sintering under reaction conditions. These findings highlight the dual role of Re and Mo in tuning both activity and stability, emphasizing the crucial influence of Mo on the long-term performance of Cu-based catalysts for CO_2_ to methanol conversion.

## 1. Introduction

The increase in greenhouse gas emissions like CO_2_, due to the continued consumption of energy through fossil fuels such as coal and fuel oil, is one of the main contributors to climate change. In this area, carbon capture, storage and utilization (CCSU) technologies have gained great interest. The use of CO_2_ and renewable energy for the synthesis of high value-added chemicals and fuels has gained great interest [1,2]. In particular, the catalytic hydrogenation of CO_2_ to methanol is highly interesting, since methanol is a promising energy vector for a sustainable economy [3] and one of the most important feedstocks for the chemical industry. Currently, methanol is obtained from synthesis gas, coming from fossil fuels such as natural gas and coal [4,5]. Thus, the use of CO_2_ as a C_1_ building block and hydrogen, generated through renewable sources like water electrolysis, is a sustainable route for CO_2_ recycling and hydrogen storage. Many catalytic processes, including photo- [6,7,8], electro- [9] and thermo-catalytic CO_2_ reduction, have been investigated in recent years, being the last ones the more mature route. Nevertheless, the direct thermo-catalytic CO_2_ hydrogenation to produce methanol is a major challenge due to the chemical inertness of the CO_2_ molecule, (∆G^0^_298K_ = 396 kJ/mol) along with the thermodynamic limitations of the reaction and the competitive reverse water gas shift reaction (RWGS) to CO [10,11,12]. The most studied catalysts for methanol synthesis from CO_2_ and H_2_ are those based on copper–zinc oxide because they show the best methanol formation yields; however, catalyst stability is still a critical point. Catalyst deactivation has been ascribed to metal sintering which is accelerated in the presence of the reaction generated by water. In this context, metal oxide promoters have been used to increase not only the stability of Cu/ZnO-based catalysts but also the activity and selectivity to methanol [13,14]. In this regard, the addition of Al_2_O_3_ in the commercial Cu/ZnO/Al_2_O_3_ formulation acts as a structural promoter, improving the dispersion of copper–zinc oxide nanoparticles and enhancing the thermal stability of the catalyst. In addition, other promoters such as Ga_2_O_3_, CeO_2_, Cr_2_O_3_, MgO, WO_3_, etc. have been studied [15,16,17,18,19,20,21,22,23]. For example, it has been proposed that adding a small amount of WO_3_ (2–5 at%) on a Cu-ZnO-ZrO_2_ catalyst increases both the activity and methanol selectivity [24]. This has been related to an increase in the dispersion of the active components and of the surface area of the catalyst, together with an increase in the number of basic sites and the reducibility of the catalyst [24]. The same authors performed a comparative study analyzing different transition metal oxide promoters, i.e., Cr_2_O_3_, MoO_3_ and WO_3_, on the catalytic performance of the Cu-ZnO-ZrO_2_ catalyst in the CO_2_ hydrogenation to methanol [25]. While the addition of Cr_2_O_3_ decreases the dispersion of active components, the catalyst surface area, as well as the CO_2_ adsorption capacity and reducibility, with a corresponding decrease in methanol yield, an opposite effect is observed by adding MoO_3_ and WO_3_, with methanol production being favored over WO_3_ than MoO_3_. According to these authors, methanol production is directly related to the surface area of metallic copper and the CO_2_ adsorption capacity of the catalyst. In contrast, other authors [26] did not find a promoting effect when adding MoO_3_ in a 2 wt% to a CuZnOAl_2_O_3_ catalyst. For the Cu/ZnO catalyst, Saito et al. [27] found that adding 5 wt% Cr_2_O_3_ increases the specific activity of copper by optimizing the Cu^+^/Cu^0^ ratio on the Cu surface, thus increasing the methanol yield. Also, a positive effect on stability was observed. Other promoters such as CeO_2_ and Ga_2_O_3_ have also been explored with an interesting influence on the catalytic performance [15,28,29,30]. While most of the literature studies usually focused on promoter loadings above 2 wt%, recently, it was found that the addition of ultra-low loadings of the promoter (0.01 wt%) (i.e., Ga, Ce, Zn) on Cu/SiO_2_ has an important effect on catalytic activity, explained by the stabilization of new active sites [31,32,33].

The use of ReO_3_ in Cu/ZnO catalysts in methanol synthesis has been less explored in the literature. However, it has shown interesting properties in Ni/Al_2_O_3_ and Ag/Al_2_O_3_ catalysts for improving metal dispersion, favoring H_2_ activation or hydrogen spillover and stabilizing the active site against sintering [34,35]. Accordingly, our work is focused on studying the promoting effect of this transition metal oxide on Cu/ZnO catalyst, and comparing it to the previously reported MoO_3_, paying special attention to catalyst stability and activity. Metal loadings going from ultra-low loading (0.06 wt%) to those usually reported in the literature (3.5 wt%) have been considered. Spectroscopic, catalytic and kinetic studies have been performed to identify the role of the promoter in both catalytic activity and catalyst stability. An interesting effect was found between the interaction of the Cu nanoparticle with the promoter; the copper dispersion and the water affinity of the promoter ascribed to its propensity of forming hydroxyl (Me-OH) groups, with their inhibiting effect on the sintering of the copper nanoparticles. This results in enhanced long-term catalytic activity.

## 2. Materials and Methods

### 2.1. Catalyst Synthesis

Cu/ZnO-based catalysts promoted with different amounts of transition metals (named, CuZnMe-X) were prepared by the coprecipitation method, modifying both the metal promoter (where Me is rhenium (Re) and molybdenum (Mo)) and the metal loading for each promoter (where X is between 0.06 and 3.50 wt.%). For this purpose, the same Zn:Cu ratio = 0.43 was kept constant in all cases, and the percentage of the different metal promoters was varied (Table 1).

In the synthesis of CuZnMe-X catalysts, the metal precursors employed were the following: Cu(NO_3_)_2_·3H_2_O (Sigma-Aldrich, puriss. p.a., 99–104%); (Zn(NO_3_)_2_·6H_2_O (Sigma-Aldrich, reagent grade, 98%) for the CuZn oxide reference; and NH_4_ReO_4_ (Sigma-Aldrich, >99%) or (NH_4_)_6_Mo_7_O_24_·4H_2_O (Sigma-Aldrich/Supelco, cryst. Extra pure) for the different metallic promoters.

In a typical synthesis procedure, the metallic precursors were dissolved in milliQ water to obtain a 1.1 M solution and at 0.50 mL/min (kd Scientific, KDS-200 syringe pump) into a beaker containing 200 mL of miliQ water at 65 °C, under constant stirring (350 rpm), using a 1.4 M NaOH solution to keep a constant pH of ca. 6.50. The controlled pH of 6.5 was chosen based on previous optimization studies on co-precipitated Cu/ZnO catalysts, in which this value ensures the simultaneous precipitation of Cu(II) and Zn(II) ions, while preventing the appearance of separate Cu(OH)_2_ or Zn(OH)_2_ phases [36,37,38]. The suspension was then aged under stirring at 65 °C for 2 h. The precipitate was then filtered, washed with warm deionized water to pH = 7 and dried overnight at 100 °C. The resulting dark solid was calcined in a muffle furnace as follows: from 25 to 200 °C (2 °C/min; dwell time of 1 h), from 200 to 360 °C (2 °C/min; dwell time of 1 h), and finally held at 360 °C for 9 h. After calcination, the samples were reduced in H_2_ (20 mL/min) at 200 °C for 3 h with a heating rate of 5 °C/min. The absence of residual Na in the final catalyst (Appendix A) was confirmed by XPS analysis at the Na 1s core line.

### 2.2. Catalyst Characterization

The chemical composition of unpromoted and promoted catalysts was analyzed by inductively coupled plasma optical emission spectrometry (ICP-OES) using a Varian 715-ES spectrometer (Varian, Inc., Palo Alto, CA, USA) after solid dissolution of the catalysts in an aqueous HNO_3_/HCl solution.

The surface areas of the solid samples for the different materials studied (250 mg) were calculated by applying the Brunauer–Emmett–Teller (BET) model to the interval of the N_2_ adsorption isotherm in which a linear relationship is maintained. The calcined samples were degassed in situ under vacuum at 150 °C. These isotherms were obtained from liquid nitrogen adsorption experiments at −196 °C in a Micromeritics flowsorb instrument.

X-ray diffraction patterns (XRD) were recorded with a PANalytical Cubix Pro diffractometer using monochromatic Cu Kα radiation (λ = 0.15406 nm). The average size of Cu^0^ and ZnO crystallites (JCPDS: 01-070-3038 and 01-079-0207, respectively) were calculated from the major peaks (2θ; 50.300 and 56.463°, respectively) using the Scherrer equation and assuming a shape factor k = 0.9.

High-resolution transmission electron microscopy (HRTEM) and scanning transmission electron microscopy (STEM) were carried out using a TALOS Model F200x equipment. Microanalyses were performed using an EDX Super-X system to determine the distribution of metals.

The reducibility of catalysts was determined by temperature-programmed reduction (TPR-H_2_) experiments on a Micromeritics Autochem 2910. About 50 mg of samples were initially cleaned with 30 mL/min Ar at room temperature for 30 min, and then a mixture of 10 vol% H_2_ in Ar was passed through the solid at a total flow rate of 50 mL/min, while increasing the temperature to 600 °C at a heating rate of 10 °C/min. The H_2_ consumption was measured using a thermal conductivity detector (TCD).

The surface concentration of metallic copper sites was quantified via N_2_O chemisorption followed by temperature-programmed reduction with H_2_ (TPR-H_2_), using a Micromeritics AutoChem 2910 instrument. A stoichiometric ratio of 1:2 (H_2_:Cu_S_) was assumed for H_2_ uptake. Prior to analysis, 50 mg of each CuZnMe-X catalyst were pre-reduced under a pure H_2_ flow (20 mL/min) at 200 °C for 3 h. Subsequently, the samples were purged with argon at the same temperature to remove residual hydrogen. Once cooled to 25 °C, surface oxidation of metallic Cu^0^ to Cu_2_O was carried out by exposing the samples to a 1 vol% N_2_O/He mixture (10 mL/min) for 1 h. Following this step, the samples were flushed with argon at room temperature for 15 min. The re-oxidized copper surface was then subjected to TPR-H_2_ up to 400 °C (10 vol% H_2_ in Ar, 50 mL/min) at a linear heating rate of 10 °C/min.

Temperature-programmed desorption (TPD-CO_2_) studies on in situ reduced samples of the different metal promoters were performed using a ChemStar TPx (chemisorption analyzer) instrument with a quartz reactor at atmospheric pressure, connected on-line to a mass spectrometer (MS) OMNISTARTM (GSD 320 Gas Analysis System) and a thermal conductivity detector (TCD). First, samples (ca. 50 mg diluted in 300 mg of SiC) were activated at 200 °C in a pure H_2_ flow rate of 20 mL/min for 3 h. Then, the samples were cleaned with Ar (20 mL/min) at room temperature for 30 min. After stabilization, CO_2_ was pulsed to the sample 20 times using a four-way valve (61.0 µL loop) at 50 mL/min (5% CO_2_ in Ar) for 1 h. After the adsorption step, physisorbed CO_2_ was eliminated by using Ar steam (50 mL/min) for 30 min. After that, chemisorbed CO_2_ desorption measures took place at increasing temperatures (from room temperature up to 600 °C), keeping the inert flow rate at 50 mL/min. The CO_2_ desorption was followed by MS (*m*/*z* = 44).

Surface analysis was performed by X-ray photoelectron spectroscopy (XPS) using a SPECS system equipped with a Phoibos 150 MCD-9 multichannel analyzer and a non-monochromatic Al Kα X-ray source (hυ = 1486.6 eV). Spectra were acquired at an X-ray power of 50 W, a pass energy of 30 eV, and under ultra-high vacuum (UHV) conditions (~10^−9^ mbar). Approximately 10–30 mg of the catalyst were pressed into a pellet and mounted onto a stainless-steel sample holder. Prior to XPS analysis, CuZnMe-X samples were reduced under H_2_ flow (20 mL/min) at 200 °C and atmospheric pressure for 3 h in a high-pressure reaction cell (HPCR) directly connected to the XPS chamber under UHV. Gas flows were regulated using Bronkhorst mass flow controllers. All spectra were referenced to the C1s binding energy at 284.5 eV. Data processing was carried out using the CASA XPS software package (CasaXPS Version 2.3. 16Dev52).

Raman spectroscopy was performed using a Renishaw “inVia” spectrometer coupled with an Olympus optical microscope. The setup includes a 514 nm He-Ne green laser, a 785 nm diode laser, and a CCD detector. In situ Raman measurements at 1 bar were carried out using a Linkam THMS600 cell, which provides high-precision temperature control (±0.01 °C) over a wide range (−195 to 600 °C). Typically, for in situ experiments at 1 bar, samples were first reduced ex situ under H_2_ flow (20 mL/min) and then exposed to air for 1 h. This was followed by in situ reduction in H_2_ (20 mL/min) at 200 °C. The temperature was then increased directly to 260 °C under H_2_ flow, and the system was subsequently switched to reaction conditions using a CO_2_/H_2_ mixture (1:3 molar ratio, total flow: 20 mL/min) in the temperature range of 260–280 °C. Additional in situ Raman experiments were conducted under co-feeding conditions with ca. 3.1 vol% H_2_O at room temperature and pressure, introduced via a saturator and transported by an Ar flow (20 mL/min).

Thermogravimetric analyses were performed on a Mettler-Toledo thermobalance (TGA/SDTA 851). Specifically, for each experiment, 10 mg of the samples previously calcined in air at 360 °C were heated under a synthetic air atmosphere (50 mL/min) from room temperature up to 500 °C.

### 2.3. Catalytic Test

Catalytic studies of CO_2_ hydrogenation to methanol were performed in a stainless-steel fixed-bed reactor (11 mm inner diameter and 240 mm length), equipped with a back pressure regulator (BPR, Swagelok) that allows work at a pressure range of 1–20 bar. Typically, 200 mg of catalyst (particle size 400–600 mm) were diluted in SiC at a Cat/SiC weight ratio of 0.12. Samples were reduced in situ at atmospheric pressure prior to catalytic testing (20 mL/min H_2_, 200 °C, 3 h, 10 °C/min). Constant weight hourly space velocity experiments (WHSV, 30,000 mL/h g_cat_) were performed under concentrated reaction conditions (23.7 vol% CO_2_, 71.3 vol% H_2_, 5.0 vol% N_2_) at 20 bar. Reaction temperatures ranged from 220 to 280 °C. In addition, catalytic experiments were carried out by varying the WHSV (30000–3333 mL/h g_cat_) at a constant molar ratio of H_2_/CO_2_ = 3. Finally, long-term stability studies of the different catalysts were performed under conditions of 240 °C and 20 bar for 80 h. Direct analysis of the reaction products was performed by on-line gas chromatography (GC) using Agilent-8860 equipment with TCD (PLOT/U + mol sieve column) and FID (BR-Q Plot column) detectors. Blank experiments (in the presence of SiC) showed the absence of a homogeneous contribution to the reaction.

## 3. Results and Discussion

### 3.1. Characterization of Catalysts

The structural properties of CuZnMe-X catalysts promoted with different amounts of metal were studied by powder X-ray diffraction (XRD). The diffraction patterns of the as-prepared, calcined in air, H_2_ reduced, and after-reaction samples are shown Appendix A. Characteristic peaks of CuO (at 2θ = 32.5, 35.5, 38.8, 48.7, 53.5, 58.3, 61.5, 66.2, 67.9, 68.1, 72.4, 75.2°) and ZnO (at 2θ = 31.7, 34.4, 36.2, 47.5, 56.6, 62.8, 67.9, 69.1, 76.9°) phases were observed in the diffraction patterns of the as-prepared and calcined materials (Appendix A, respectively), whereas, in the reduced materials, the characteristic peaks of the Cu^0^ (at 2θ = 43.2, 50.4, 74.0°) and ZnO phases were observed. However, no characteristic diffraction peaks of the rhenium and molybdenum phases were observed in any case, suggesting high dispersion of the metal promoter in all samples. This is confirmed by HRTEM and STEM-EDX mapping performed on ex situ reduced catalysts. As shown in Figure 1, independently of the promoter loading, molybdenum is well-dispersed over the sample in close contact with Cu and ZnO, whereas rhenium seems to interact preferentially with ZnO. In addition, similar distributions of the promoters have been found on spent samples (Appendix A), indicating no significant changes under reaction conditions.

The average particle size of Cu^0^ and ZnO were measured from the XRD patterns using Scherrer’s equation analysis (Table 1). It was shown that, independently on the metal loading, the addition of MoO_3_ results in a lower Cu particle size (around 8 nm) compared to that of Re (around 17 nm) and the un-promoted Cu/ZnO sample (23 nm). In addition, MoO_3_ favors a higher surface area of the catalyst (ca. 25 m^2^/g) compared to that of the un-promoted Cu/ZnO sample (10 m^2^/g) or CuZnRe-0.06 catalyst (ca. 9 m^2^/g). The amount of surface copper atoms determined from N_2_O chemisorption analysis does not follow the expected trend based on metal dispersion, being markedly lower for Mo-based catalysts, indicating partial blocking of surface copper atoms by either MoO_X_ or ZnOx. In this direction, the XPS data in Table 1 shows a Cu/Zn ratio between 0.8 and 1 in all reduced catalysts, which is lower than the theoretical 2.4 associated with partial migration of ZnOx over the Cu nanoparticle, a trend already reported in the literature [39]. Furthermore, a marked decrease in the surface atomic Cu/Mo ratio of the MoO_3_-promoted samples is observed, specifically at high Mo loadings (with Cu/Mo of 5.02 and 0.80 in CuZnMo-0.44 and CuZnMo-3.5, respectively), compared to the Cu/Re ratio of the ReO_3_ promoted samples (with Cu/Re of 32.8 and 29.6 in CuZnRe-0.50 and CuZnRe-3.5, respectively). This is in line with the above reported STEM-EDX data, where Re atoms were mainly located close to ZnO, whereas Mo atoms were located close to Cu and ZnO. The XPS binding energies (BE, eV) of the Cu 2*p*_3/2_ and Zn 2*p*_3/2_ core lines (Appendix A) in the H_2_-reduced samples corresponds in all cases to Cu^0^ (Cu 2*p*_3/2_ ~932.6 ± 0.1 eV and CuL_3_M_45_M_45_ 918.4 eV) and Zn^2+^ in ZnO (Zn 2*p*_3/2_ ~1022.0 ± 0.3 eV and ZnL_3_M_45_M_45_ 987.8 eV). No shoulder is observed at KE 991.6 eV in the ZnL_3_M_45_M_45_ Auger line, discarding the possible formation of CuZn alloys [40,41]. A detailed analysis at the Cu AES peak shows a component at KE at ~916 eV in all promoted samples (Appendix A), while it is hardly observed on the un-promoted sample (Appendix A), accounting for the stabilization of Cu^+^ species in the presence of the promoter. The contribution of Cu^+^ species seems to be highest on the samples with low promoter loadings (0.06 wt%), specifically in the presence of rhenium (see Table 1). However, the low transmission of the samples in IR spectroscopic studies due to its black color impedes the identification of Cu^+^ species by IR of CO as a probe molecule.

Finally, in the case of the Mo 3*d* core line (Appendix A), a shift is observed in the BE of Mo 3*d*_5/2_ to lower values (from 233.3 to 232.6 eV) with increasing Mo loading from 0.06 to 3.5 wt%. This corresponds to a change in the dispersion and local environment of Mo^6+^ species from highly dispersed molybdenum oxide species in a tetrahedral coordination to a more MoO_3_ like state with octahedral coordination [42,43,44,45,46]. In addition, a new component at low BE, 230.8 eV, corresponding to Mo sites in a lower oxidation state (5+), appears at increasing Mo loadings. In the case of Re 4*f*_7/2_ (Appendix A), an opposite effect is observed, with a shift to higher BE from 43.3 to 44.2 eV with increasing Re loading (from 0.5 to 3.5 wt%), which may be related to changes in the oxidation state (4+ and 6+, respectively). This indicates a more defective state of rhenium at low loadings, while it becomes more oxidized with increasing rhenium loading. This different trend in the chemical states of the metal oxide promoter can be associated with their different location, either on the Cu NP as in the case of Mo, which favor a more reduced state due to H_2_ spillover, or on the ZnO particles as in the case of Re, which favor a more oxidized state.

The reducibility of all samples calcined in air at 360 °C was studied by temperature-programmed reduction (H_2_-TPR). The results are shown in Figure 2. A broad reduction peak and a shift in the maxima reduction temperature to higher values were observed, specifically on the molybdenum-promoted catalysts (Figure 2, patterns f and g), indicating a copper–molybdenum interaction in line with EDX analysis.

The Raman spectra of calcined and in situ reduced samples are displayed in Figure 3. In all calcined samples (Figure 3A,B), a similar pattern is observed with bands at ca. 280, 337 and 630 cm^−1^, characteristic of CuO [47,48] (see CuO reference in Appendix A), and at ca. 580 cm^−1^, corresponding to partially reduced copper species [49,50,51].

In MoO_3_-promoted samples, additional bands appear in the 800–900 cm^−1^ region, assigned to the asymmetric vibrational modes of bridging Mo-O-Mo and/or O-Mo-O bonds, and in the 900–1000 cm^−1^ range, associated with the symmetric and asymmetric stretching modes of terminal Mo=O bonds, which correspond to tetrahedral and/or octahedral Mo species on the surface [52,53]. The intensity of these bands increases with increasing Mo loading. However, in the case of ReO_3_-promoted samples, the weak and broad Raman features observed between 600 and 1000 cm^−1^ are consistent with disorder-induced scattering in ReO_3_, whose ideal cubic structure is Raman-inactive. These bands arise from oxygen vacancies and surface Re=O/ReO_4_^−^ species formed upon mild oxidation [54,55].

Upon in situ reduction in H_2_ at 200 °C (Figure 3C,D), distinct differences emerge in the Raman spectra of the MoO_3_ and ReO_3_-promoted catalysts. In the MoO_3_-containing samples, a marked decrease in the intensity of the Cu_2_O- and CuO-related bands is observed. Additionally, two weak bands appear in the 430–490 cm^−1^ region, assigned to Cu-OH species [49] (see Cu(OH)_2_ reference in Appendix A), likely originating due to water generated during the reduction process. Notoriously, no Raman bands related to Cu species are observed at higher Mo loadings (CuZnMo-3.50), and only the Raman bands of molybdenum species are observed. This is in line with XPS and N_2_O titration data where a partial blocking of surface copper sites is observed with increasing Mo loading under reducing conditions. In contrast, in the ReO_3_-promoted samples, an intense Raman band is observed at 560 cm^−1^, with a shoulder at 440 cm^−1^ (associated with reduced copper phases (Cu_2_O) and Cu(OH)_X_, respectively), which remains unaltered even at higher Re loadings. Regarding the nature of the promoter, in the case of the CuZnMo-3.5 sample, the initial components at 820 and 931 cm^−1^ decrease in intensity under in situ H_2_ conditions, leading to the formation of a single component at 889 cm^−1^, which is associated with Mo-O vibrations of shear defects in MoO_3−X_ [56,57,58,59].

Finally, the Raman spectra of all catalysts under in situ reaction conditions (CO_2_/H_2_, 1/3 molar ratio, 260 °C) are comparatively presented in Appendix A. In principle, no substantial spectral changes are detected compared to the corresponding in situ reduced states, indicating that the structural features of the active phases remain largely unaltered under the applied reaction conditions. However, in the CuZnMo-3.50 sample (Appendix A), a broadening of the Raman band is observed at 889 cm^−1^, with a shoulder at 848 cm^−1^, which could be related to the formation of Mo-OH-like species [56].

Moreover, the surface basicity of reduced catalysts was studied by CO_2_ temperature-programmed desorption (CO_2_-TPD). Appendix A shows three distinct CO_2_ desorption regions: below 200 °C, between 200 and 500 °C, and above 500 °C. The intermediate region (200–500 °C) is considered the catalytically relevant zone, as several authors have reported that a higher density of basic sites in this region is crucial for stabilizing reaction intermediates that enhance methanol selectivity [15,60]. This region is mainly associated with surface defects, low-coordination oxygen anions, and metal–oxygen pair sites [61,62]. In our study, minor variations are observed within the range of 200–500 °C, with a slightly higher desorption temperature in the Mo-promoted samples.

### 3.2. Catalytic Studies

The catalytic performance of the CuZnMe-X catalysts in CO_2_ hydrogenation at 20 bar, time–space velocity (WHSV) of 30,000 mL/h g_cat_ and in the temperature range between 220 and 280 °C is given in Appendix A. As a reference, an un-promoted CuZn-based catalyst synthesized by the same method was used. In all cases, CO_2_ conversion as well as methanol production (expressed as g_MeOH_/h g_cat_) increase with reaction temperature, while the selectivity to methanol decreases at the expense of CO formation, which is consistent with the endothermic nature of the competing reverse water–gas shift (RWGS) secondary reaction. Compared to the un-promoted CuZn catalyst, and independent of the reaction temperature, the methanol production normalized to grams of the catalyst enhances for both CuZnMo and CuZnRe catalysts at low promoter loadings (0.06 and 0.5 wt%, respectively), while it is reduced at high promoter loading (i.e., 3.5 wt%) (Appendix A).

However, as reported in Table 1, the Mo promoter increases the surface area of the catalyst; thus, to more accurately assign activity to surfacedistributed centers, methanol production was normalized by the surface area of each catalyst (m^2^/g_cat_). Under these conditions, as before, both the CuZnRe-0.06 and CuZnRe-0.5 catalysts show enhanced methanol production (ca. 0.052 g_MeOH_/h m^2^_cat_ at 240 °C) compared to that of the un-promoted CuZn catalyst (0.042 g_MeOH_/h m^2^_cat_) (Appendix A). In contrast, the CuZnMo-0.06 catalyst shows slightly lower methanol production (0.039 g_MeOH_/h m^2^_cat_ at 240 °C) compared to the CuZn catalyst (0.042 g_MeOH_/h m^2^_cat_), being lower in the rest of the Mo- promoted catalysts. Consequently, the order of methanol production per catalyst surface area at 240 °C is CuZnRe-0.06 ≈ CuZnRe-0.5 > CuZn ≥ CuZnMo-0.06 ≈ CuZnRe-3.5 > CuZnMo-0.5 > CuZnMo-3.5. A similar trend is also observed in CO production, where the methanol versus CO production is higher in the Re-promoted samples than in the Mo-based catalysts (see Appendix A). CO comes from the reverse gas shift reaction (RWGS) which is considered a structure-sensitive reaction, influenced by Cu particle size [63], while the formation of additional Cu-ZnO-MeO_3_ interfacial sites providing additional sites for CO as well as for methanol formation cannot be excluded.

To determine the intrinsic activity of active sites, initial reaction rates of methanol and CO formation have been calculated at 240 °C and 20 bar (Appendix A) and compiled in Figure 4. These initial rates have been determined by calculating the methanol and CO production extrapolated at zero contact time (see Appendix A). From these results, a volcano-like tendency is observed between the methanol and CO initial rate (g_MeOH_/h m^2^_cat_ and g_CO_/h m^2^_cat_) and ReO_3_ and MoO_3_ loading, with a maximum value at ultra-low promoter loadings, which is independent of the reaction temperature. These results indicate clear differences between active centers and a drastic reduction in their intrinsic activities at increasing the amount of the promoter. Moreover, it is clearly seen that the methanol formation rate at a constant molar number (mmol_Me_/g_cat_) is higher in ReO_3_-promoted catalysts than in the MoO_3_-promoted ones.

Furthermore, the ratio between the initial rate of methanol formation versus CO is similar on all catalysts, around 9, except in the CuZnMo-3.5 sample with a value of 6.9, being slightly lower in all cases compared to that of the un-promoted CuZn catalyst (10.9) (Appendix A). The similarity in these trends and the nearly constant MeOH/CO ratio suggest that both products share a common active site (see more details in Appendix A).

The negative effect of the promoter at higher loadings, which is more pronounced in Mo-containing catalysts, is in line with the lower amount of copper atoms on the catalyst surface, as determined by N_2_O chemisorption analysis (Table 1), due to partial blocking of the copper particle by some promoter or ZnOx layers, as evidenced by XPS studies (Table 1 and Appendix A). To remove this effect and determine the intrinsic activity of surfaceexposed copper sites, the initial rates of methanol and CO production have been normalized to the number of surface copper atoms determined by N_2_O titration (Appendix A). In this case, a similar volcano trend as in Figure 4 is observed (Appendix A). Hence, Re-promoted catalysts exhibit a high initial intrinsic methanol rate per exposed Cu atom at ultra-low loadings, decreasing to an almost constant value at promoter loadings above 0.3 mmol/g_cat_, highlighting an electronic or structural effect localized at the active site under ultra-low Re loading. In contrast, Mo-promoted catalysts show significant variations in Cu-site reactivity, indicating that Mo negatively alters the intrinsic nature of the active site while also promoting partial blocking of surface copper atoms at higher loadings.

Next, in Figure 5, the variation in the selectivity to methanol with the CO_2_ conversion is compared. All Re-promoted samples show a similar trend in methanol selectivity, being slightly lower than that of the un-promoted CuZn sample (Figure 5A). However, major discrepancies are observed in the Mo-based samples (Figure 5B). This may be due to the different interaction of the promoter with the copper nanoparticle, as determined from STEM-EDX, affecting in major extend the nature of active sites in the Mo-based catalysts and resulting in significant variations in the reactivity of Cu sites, as discussed above. Hence, the lowest methanol selectivity is observed in the CuZnMo-3.5 sample. This sample displays heterogeneity in Mo sites, with stabilization of Mo^5+^ and Mo^6+^, and surface enrichment of Mo, which explains this different catalytic trend.

The previously described trends in methanol and CO production and selectivity are consistent with the differences observed in their apparent activation energies for methanol and CO formation (Appendix A). The addition of Re significantly decreases the apparent activation energy for methanol synthesis from 22 kJ/mol to approximately 14 kJ/mol. In the case of Mo, at low promoter loadings, the apparent activation energy for methanol formation is around 14.8 kJ/mol, but it increases almost linearly with Mo loading, reaching values close to 36.8 kJ/mol for the CuZnMo-3.5 sample. For CO formation, the apparent activation energy is consistently lower for all Re-promoted catalysts (~88 kJ/mol) compared to the CuZn reference (92.4 kJ/mol), whereas it is higher for Mo-containing samples (97.4–101.8 kJ/mol). These observations clearly reveal promoter-dependent differences in the nature of the active sites, the reaction mechanism and/or the adsorption enthalpy of key intermediates. The enhanced methanol formation observed at ultra-low Re and Mo loadings can be ascribed to the stabilization of Cu_2_O species, as confirmed by in situ Raman and XPS, reducing the energy barriers of methanol production. For Mo-promoted catalysts, the progressive increase in the apparent activation energy for methanol formation with Mo content correlates with the generation of structural defects and enhanced Cu-MoO_3_ interactions that modify the electronic properties of the copper sites leading to decreased methanol selectivity.

### 3.3. Stability Studies

An additional key aspect of Cu/ZnO catalysts is their stability under long-term reaction conditions. Therefore, long-term catalytic stability studies of selected CuZnMe-X catalysts were carried out at 240 °C, 20 bar and for 80 h and compared them with that of the un-promoted CuZn reference catalyst. As shown in Figure 6, the largest decrease in methanol production happens during the first 20 h, while from that time it practically stabilizes, with all samples showing the same catalytic trend with time on stream. Moreover, it is observed that the reference CuZn sample is the one that suffers the strongest deactivation, following the trend CuZn > CuZnRe-0.50 > CuZnMo-0.06 > CuZnMo-0.44 > CuZnMo-3.50. From these data, it seems clear that the addition of the different metallic promoters causes a notorious positive effect on the stability of the catalyst in methanol synthesis. The deactivation constants evaluated on selected catalysts after 20 and 80 h are given in Table 2 and Appendix A. There, at 20 and 80 h, the lowest deactivation constant is observed for CuZnMo-0.44 (*K* = 0.030 and 0.044, respectively) and CuZnMo-3.50 (*K* = 0.016 and 0.030, respectively), with significantly higher values for the reference CuZn catalyst (*K* = 0.164 and 0.168, respectively). Thus, after 80 h of reaction, the stability of CuZnMo-0.44 and CuZnMo-3.50 catalysts improved by 73.8% and 82.1%, respectively, with respect to un-promoted CuZn. Furthermore, a comparison with the commercial CuZnOAl_2_O_3_ catalyst (Appendix A) reveals comparable or even superior stability, despite the markedly lower surface areas (8–28 vs. 60–80 m^2^/g_cat_). This finding indicates that Re and Mo promoters improve the efficiency and resilience of the active sites, extending beyond the mere structural contribution of high surface areas.

Sintering of copper and zinc components has been widely reported as responsible for catalyst deactivation in Cu/ZnO catalysts in the methanol synthesis. In line with this, a clear increase in the size of Cu^0^ nanoparticles (about 40–55%) and, to a lesser extent, of ZnO nanoparticles (about 2–14%) is observed in all cases after 80 h catalytic stability studies, except in the most stable CuZnMo-0.44 and CuZnMo-3.50 samples, where the Cu particle size only increases by 38% and 20%, respectively (Table 2). This could be related to the increased copper dispersion favored by the larger catalyst surface area and smaller Cu particle size and the higher interaction of the Cu nanoparticle with MoO_3_, reducing Cu sintering. Another variable worth considering is the higher water affinity of MoO_3_ in trapping water from the catalyst surface in the form of Mo-OH moieties, compared with the higher hydrophobicity and lower water affinity of ReO_3_. TG analysis performed on calcined samples shows a higher water accumulation in the CuZnMo-0.44 and CuZnMo-3.50 samples (see Appendix A), both showing enhanced stability. This could be related to the presence of Mo=O moieties that, in the presence of water, converted into Mo-OH, as demonstrated by in situ Raman studies performed on the CuZnMo-3.50 sample by co-adding ~3.1 vol% H_2_O/Ar (Appendix A).

## 4. Conclusions

This work demonstrates that the addition of MoO_3_ and ReO_3_ to Cu/ZnO catalysts strongly influences the physical and chemical properties of the catalyst and its performance in the CO_2_ hydrogenation to methanol. Both promoters show a volcano-type relationship, with ultra-low loadings (~0.06 wt%) maximizing CO_2_ conversion and methanol productivity, while at increasing promoter loading (to 3.5 wt%), the catalytic activity for CO_2_ conversion and the methanol yield tended to decrease. The enhanced performance at low loadings arises from the stabilization of Cu_2_O species and the creation of new interfacial sites at the metal–promoter boundary, which lower the apparent activation energy for methanol formation.

At higher promoter contents, partial coverage of Cu nanoparticles by MeOx or ZnO_X_ layers reduces the number of exposed Cu sites, leading to decreased activity. Re is preferentially stabilized on ZnO and maintains Cu_2_O species under reaction conditions, whereas in the case of Mo-based catalysts additional effects at increasing Mo loadings have been observed increasing the apparent activation energy of methanol formation. This includes the coexistence of Mo^6+^, Mo^5+^ species, a large covering of the Cu NP with MoOx domains and a strong Cu-MoO_X_ interaction. As a consequence, the overall decrease in activity as well as the methanol selectivity is due to CO formation.

Besides activity, catalyst stability is a key point in CO_2_-to-methanol conversion technology. Long-term catalytic studies performed up to 80 h showed that the Cu/ZnO-based reference sample underwent strong deactivation. However, the addition of molybdenum plays a key role in inhibiting catalyst deactivation. Specifically, samples CuZnMo-0.44 and CuZnMo-3.50 showed markedly lower deactivation compared to the reference CuZn catalyst. In addition to the interaction of the Cu nanoparticle with MoO_3_ and a higher Cu dispersion, the promoter’s affinity for water was proposed to play a key role in trapping water in a chemical bond, avoiding strong sintering of the copper particles, which is one of the main reasons for catalyst deactivation.

## Figures and Tables

**Figure 1 nanomaterials-15-01730-f001:**
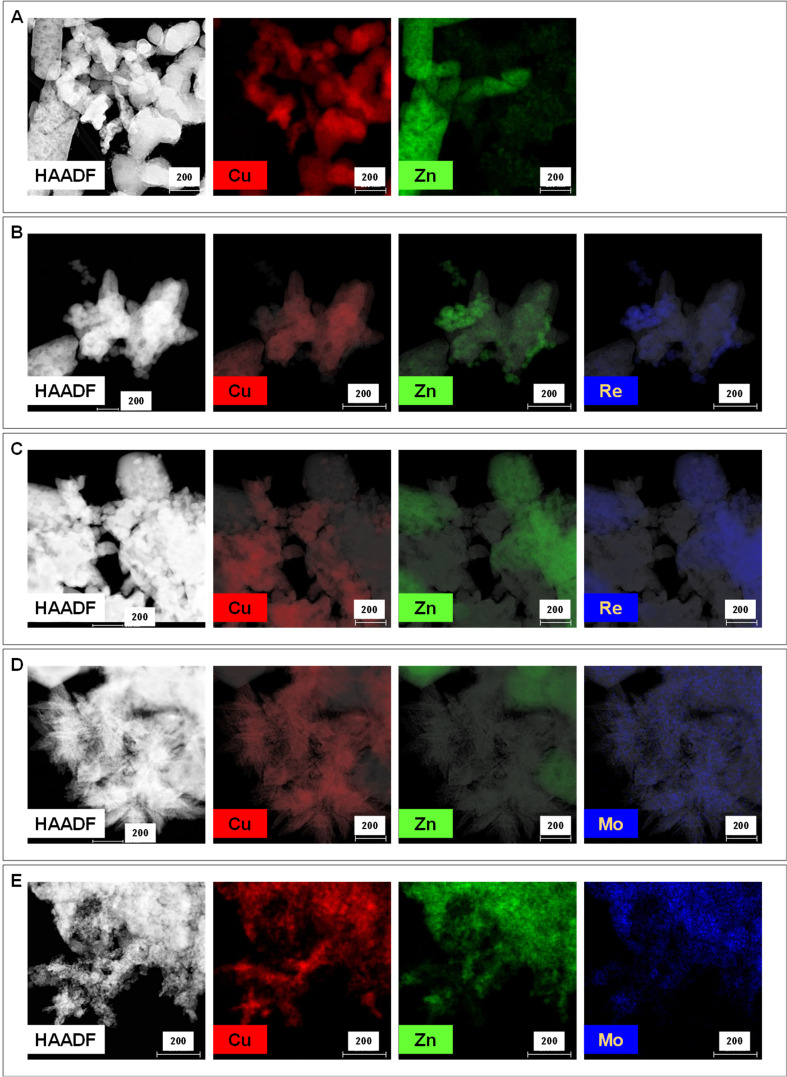
STEM-EDX microscopy studies on reduced CuZnMe-X catalysts: (**A**) CuZn; (**B**) CuZnRe-0.5; (**C**) CuZnRe-3.5; (**D**) CuZnMo-0.44; and (**E**) CuZnMo-3.5.

**Figure 2 nanomaterials-15-01730-f002:**
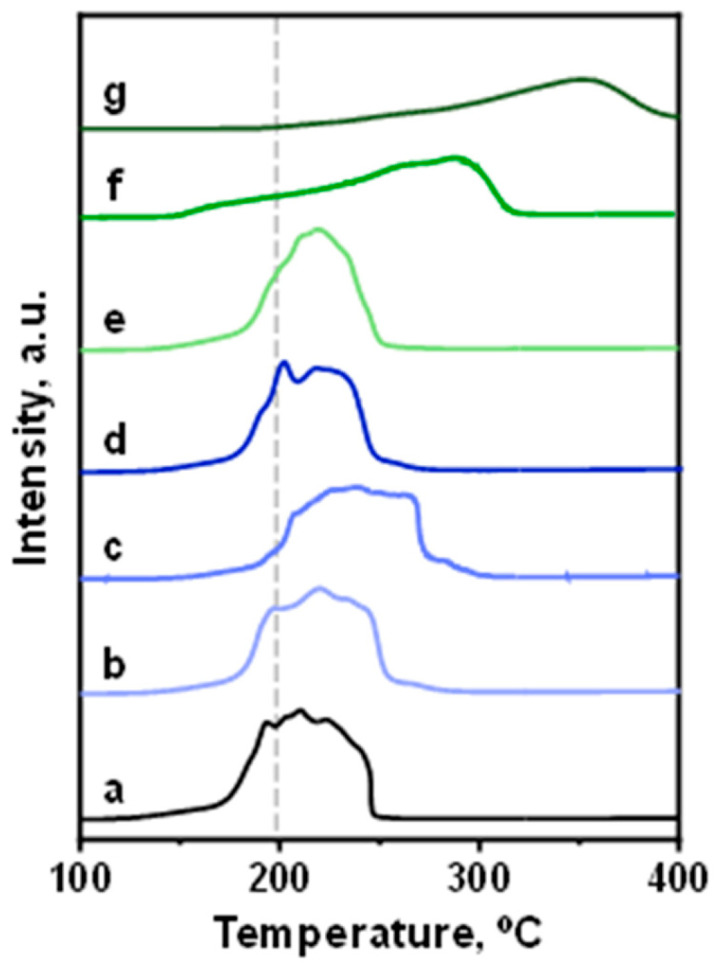
Temperature-programmed reduction (H_2_-TPR) studies in 10% H_2_/Ar flow on CuZnMe-X catalyst: (a) CuZn; (b) CuZnRe-0.06; (c) CuZnRe-0.50; (d) CuZnRe-3.50; (e) CuZnMo-0.06; (f) CuZnMo-0.44; and (g) CuZnMo-3.50.

**Figure 3 nanomaterials-15-01730-f003:**
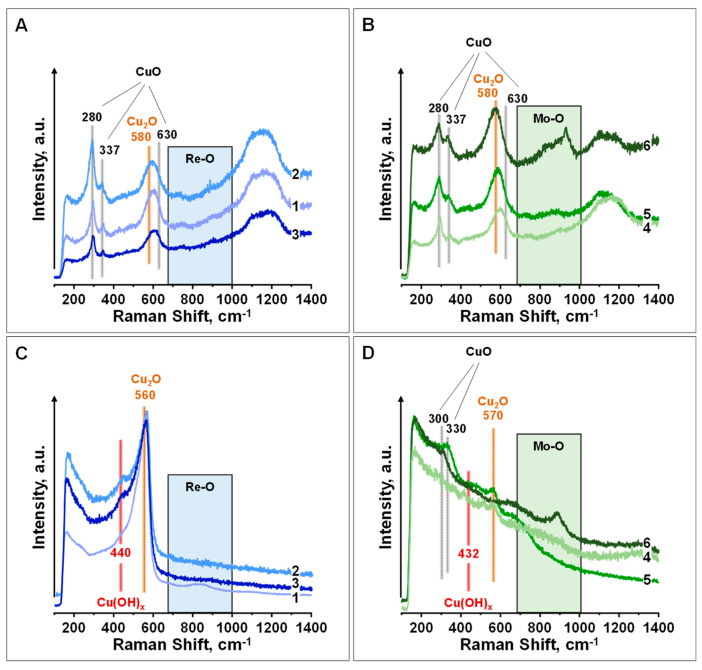
In situ Raman spectroscopic studies on CuZnMe-X catalysts (recorded with 785 nm diode laser): (**A**,**C**) Raman spectra of rhenium-promoted samples and (**B**,**D**) Raman spectra of molybdenum-promoted samples. (**A**,**B**) Spectra of calcined samples, in air at 360 °C (above); (**C**,**D**) spectra of samples after in situ reduction in H_2_ at 200 °C (below). Catalysts: (-1-) CuZnRe-0.06; (-2-) CuZnRe-0.50; (-3-) CuZnRe-3.50; (-4-) CuZnMo-0.06; (-5-) CuZnMo-0.44; and (-6-) CuZnMo-3.50.

**Figure 4 nanomaterials-15-01730-f004:**
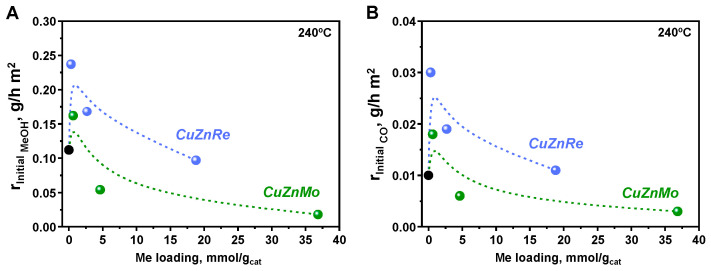
Initial reaction rate of (**A**) methanol and (**B**) CO formation expressed per surface area as a function of Re and Mo molar loading on CuZnRe-X and CuZnMo-X catalysts, respectively, at 240 °C. Reaction conditions: H_2_/CO_2_ = 3, 20 bar and a WHSV of 30000–3333 mL/h g_cat_. The black dots represent the un-promoted CuZn sample.

**Figure 5 nanomaterials-15-01730-f005:**
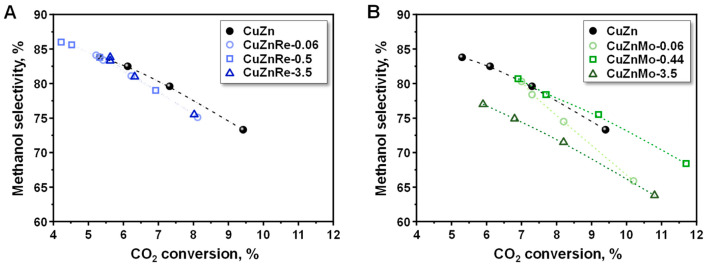
Methanol selectivity as a function of CO_2_ conversion for (**A**) CuZnRe-X and (**B**) CuZnMo-X catalysts in the CO_2_ hydrogenation reaction at 20 bar. Reaction conditions: H_2_/CO_2_ = 3, 240 °C, WHSV = 30000–3333 mL/h g_cat_.

**Figure 6 nanomaterials-15-01730-f006:**
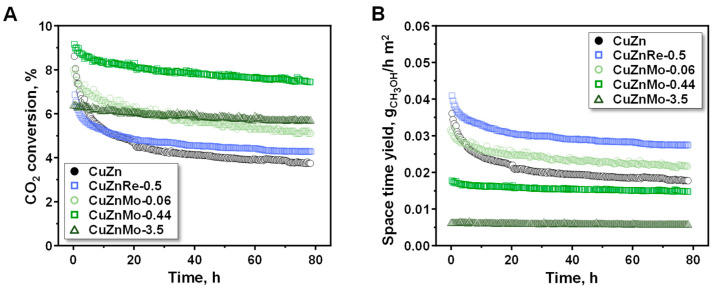
Catalytic performance of CuZnMe-X-based catalysts in the CO_2_ hydrogenation at 20 bar and 240 °C: (**A**) evolution of conversion of CO_2_ with time on stream; and (**B**) evolution of methanol productivity, space time yield (in g_CH3OH_/h m^2^), with time on stream.

**Table 1 nanomaterials-15-01730-t001:** Physico-chemical properties of un-promoted and Me-promoted CuZnMe-X (Me = Mo, Re) catalysts.

Catalyst	Ratio Cu/Me ^a^	BET Surface Area (m^2^/g) ^b^	Crystallite Size, nm ^c^		H_2_-TPR	N_2_O-TPR ^e^		XPS ^f^		
			(2 0 0) Facet of Cu^0^	(1 1 0) Facet of ZnO	Maximum Temperature (°C) ^d^	µmol Cu_S_/g_cat_	µmol Cu_S_/m^2^	Cu/Zn Ratio	Cu/Me Ratio	Cu^+^/Cu^0^ Ratio ^f1^
CuZn	--	10.1	23.1	43.8	210.4	1006	99.6	0.87	--	0.149
CuZnRe-0.06	3281	9.2	17.0	40.1	220.2	1730	189.1	0.90	--	0.237
CuZnRe-0.50	409	8.4	16.4	38.7	238.6	2560	303.9	0.67	32.8	0.210
CuZnRe-3.50	57	8.6	16.4	40.8	202.3	1168	135.8	0.96	29.6	0.590
CuZnMo-0.06	1690	13.7	13.7	41.0	219.1	1668	122.1	0.78	2.07	0.385
CuZnMo-0.44	239	28.4	8.3	48.8	287.7	1666	58.6	1.03	5.02	0.163
CuZnMo-3.50	29	25.0	8.4	18.6	350.8	1815	72.5	0.77	0.80	0.157

^a^ Cu/Me molar ratio in at% (Me = Re or Mo). ^b^ Samples calcined in air at 360 °C. ^c^ Samples reduced in H_2_ at 200 °C. ^d^ Temperature peak at higher signal intensity. ^e^ Cu atoms at the particle surface (Cu_S_) determined from N_2_O chemisorption data. ^f^ Surface Cu/Zn and Cu/Me atomic ratio determined by XPS, and ^f1^ the Cu^+^/Cu^0^ ratio was determined from the deconvolution of the Cu LMM Auger peak.

**Table 2 nanomaterials-15-01730-t002:** Long-term deactivation constants (*K*) and crystallite size of CuZnMe-X-based catalysts.

Catalyst ^a^	*K* ^b^ 20 h	*Ratio (K_CuZn_/K_CuZnM-X_)* 20 h	% ^c^ 20 h	*K* 80 h	*Ratio (K_CuZn_/K_CuZnM-X_)* 80 h	% ^c^ 80 h	Crystallite Size, nm ^d^		Crystallite Size, nm ^a^	
							(2 0 0) Facet of Cu^0^	(1 1 0) Facet of ZnO	(2 0 0) Facet of Cu^0^	(1 1 0) Facet of ZnO
CuZn	0.164	1.0	--	0.168	1.0	--	23.1	43.8	42.8	46.3
CuZnRe_0.50_	0.087	1.9	47.0	0.092	1.8	45.2	16.4	38.7	27.9	44.9
CuZnMo_0.06_	0.066	2.5	59.8	0.100	1.7	40.5	13.7	41.0	33.1	47.3
CuZnMo_0.44_	0.030	5.5	81.7	0.044	3.8	73.8	8.3	48.8	13.5	50.1
CuZnMo_3.50_	0.016	10.3	90.3	0.030	5.6	82.1	8.4	18.6	10.6	16.7

^a^ Samples after reaction in CO_2_/H_2_ at 240 °C and 80 h. ^b^ Deactivation constants have been calculated by a polynomial fit of order 1 with respect to Ln (X-CO_2(t=0)_/X-CO_2(t=x)_) versus time. ^c^ Percentage of deactivation relative to the unpromoted CuZn catalyst. ^d^ Samples reduced in H_2_ at 200 °C.

## Data Availability

All data are available by request to the corresponding authors.

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
