# Peer review of "Enhancing Stability of Cu/ZnO Catalysts in the CO2 Hydrogenation to Methanol by the Addition of MoO3 and ReO3 Promoters"

_nanomaterials, 2025, doi:10.3390/nano15221730_

Round 1

Reviewer 1 Report

Comments and Suggestions for Authors

This manuscript presents a comprehensive investigation into the promotional effects of MoO3 and ReO3 on Cu/ZnO catalysts for COâ‚‚ hydrogenation to methanol, with particular emphasis on catalyst stability. The study employs a range of characterization techniques and catalytic tests to demonstrate that ultralow loadings of both promoters enhance methanol productivity, while MoO3 significantly improves long-term stability. However, several key issues require clarification and additional analysis to strengthen the conclusions. Major revisions are recommended, as detailed below:

  1. The in-situ Raman spectrum of the CuZnRe-0.50 catalyst (Fig. 3c) is notably different, lacking the characteristic Cu2O peak at ~560 cm-1 observed in other Re-promoted samples. A convincing explanation for this distinct behavior is needed.
  2. A mechanistic interpretation should be provided to reconcile the observation that the initial CO formation rate is higher than that of methanol (Fig.4), while methanol becomes the dominant product in Fig.5.
  3. The reported Mo 3d5/2 binding energy of 233.3 eV for the low Mo-loading catalyst (line 4) is atypically high compared to literature values. The origin of this positive shift requires clarification.
  4. While the manuscript qualitatively discusses Re stabilizing Cu+ and Mo enhancing Cu–Mo interaction, a quantitative analysis of the Cu0/Cu+ active site ratio via XPS curve fitting is necessary to substantiate these claims.
  5. The slightly lower methanol selectivity of Re-promoted catalysts compared to un-promoted CuZn remains unexplained (Fig.5). Furthermore, during the 80 h test, while the CO2 conversion over the Re-promoted catalyst was only marginally higher than that of the un-promoted CuZn, its reported methanol space-time yield was substantially greater. Potential reasons should be discussed.
  6. The discussion on long-term stability and anti-sintering effects would be strengthened by directly referencing and discussing the spent catalyst XRD patterns in Fig. S1D. Additionally, the chemical state of Cu after the 80 h stability test should be examined to determine if deactivation involves changes in Cu speciation.
  7. The introduction mentions conventional promoters like Al2O3 and Ga2O3 but lacks a comparative discussion highlighting the specific advantages of MoO3. Such a comparison would better underscore the novelty of the work.
  8. The manuscript focuses solely on thermal catalysis but does not connect to broader CO2 reduction fields (e.g., electrocatalysis, photocatalysis). Please cite representative works (e.g., Nano Res., 2024, 17, 7194; Chemical Engineering Journal, 2024, 498, 155576) to situate this study within the wider CO2 utilization landscape, improving the manuscript’s academic breadth.

Reviewer 2 Report

Comments and Suggestions for Authors

This article systematically investigated the effects of two additives, MoO3 and ReO3, on the activity and stability of the catalyst for the hydrogenation of CO2 to methanol. The study highlighted the dual roles of Mo and Re in regulating activity and stability, and emphasized that Mo has a crucial impact on the long-term performance of copper-based catalysts in the process of converting CO2 to methanol.

  1. In the Introduction section, the content of the first paragraph is too lengthy. It is suggested that the author divide it into several paragraphs in order to enhance the logicality and readability of the article.
  2. In the synthesis of the catalyst, could the authors briefly explain the basis for choosing the controlled solution pH of 6.5? If the pH value deviates from this pH, what impact will it have on the synthesis of the catalyst? By regulating the pH with NaOH, will there be residual Na in the catalyst, thereby affecting the catalytic performance?
  3. In the supplementary document, it is suggested that the abbreviations of the titles be standardized in the figures (for example, Figures S11 and S12), as well as the size of the figures and the size of the text within the figures (e.g., Figure S9) be unified.
  4. It is suggested to supplement HRTEM images of representative catalyst before and after the long-term stability reaction, so that better understand the mechanism of the catalyst's resistance to sintering from an atomic scale perspective.
  5. In the stability test, Mo can increase the conversion rate of CO2, while the introduction of Re significantly enhances the yield of methanol. If Cu/ZnO catalyst is loaded with a certain amount of Mo and Re simultaneously, will it be possible to achieve an increase in the CO2 conversion rate while also improving the selectivity of methanol?
  6. It is suggested that the author should further enrich the research on the advancements in the field of methanol synthesis in recent years in the Introduction section. For instance, some recent published works ("Acta Phys. -Chim. Sin. 2023, 39(11), 2212042.", "Nano Research Energy 2024, 3: e9120112", and "Nano-Micro Lett. (2020) 12:18") might be helpful for the article.

Reviewer 3 Report

Comments and Suggestions for Authors

The manuscript reports two sets of Cu/ZnO catalsyts, each promoted with MoO3 or ReO3. These samples were characterized in detail for their physic propertities and for their catalytic activity in CO2 hydrogenation. The results suggest that an optimum metal loading may exist for both sets of catalysts. Although it provides a lot of data, the scientific value of the data and the conclusions drawn from the data are very limited. Also, the manuscript is very long. Perhaps a much more concise version of the manuscript can be considered for publication.  

Here are a few specifica comments.

(1) The introduction should focus on the background why MoO3 and ReO3 were selected for this studay. What was stated now, "MoO3 and ReO3 being less explored in the literature", would be a very weak motivation.

(2) How were the "initial reaction rates" defined? This should be provided in the manuscript. 

(3) There were many seemingly "repeatative" paragraphs that interpreate the same set of catalytic activity data, but the conclusions drawn in each paragraph appear to be controdicting. For example, the first paragraph under 3.2 provide the ranking in activity of the catalysts. Then, in the next paragraph, the data were re-interpretated as "initial reaction rates". As another example, on Page 11, it concluded that "The similarity in these trends and the nearly constandt MeOH/CO ration suggest that both products orginated from conmmon active sites". Because all catalysts (excep Mo3.5) showed the same value, couldn't one further conclude that the nature of active sites are the same for all the catalysts. Then what were the point for the following paragraphes to argue the actives sites are different from catalysts to catalysts. Doesn't Figure 5 suppor the same coclusion already stated. How do the data lead to conlusion that "the nature of the active sites, the reaction mechamism..." are different? 

(4) Although the durability of the MoO3 and ReO3 promoted catalysts appear better than the Cu/Zn. Were they better than other catalsyts well known in the literature, such as Cu/Zn/Al or Cu/Zn/Zr? 

(5) What do it mean in the conclusion statement that "the addtion of MoO3 and ReO3 to Cu/ZnO catalysts strongly influences the structure of the catalysts"? Physic properties? chemcial structures?

Round 2

Reviewer 1 Report

Comments and Suggestions for Authors

The authors have revised the manuscript according the reviewer's comments and now its ok for accepted as current version

Reviewer 2 Report

Comments and Suggestions for Authors

The authors have revised the manuscript according to the comments from Reviewers. It could be accepted for publication.

Reviewer 3 Report

Comments and Suggestions for Authors

The authors have address most of the comments that I raised. The paper can be considered for publication.